# Men's education and intimate partner violence—Beyond the victim-oriented perspective: Evidence from demographic and health surveys in Central Africa

Zacharie Tsala Dimbuene[1]*, Bright Opoku Ahinkorah[2], Dickson Abanimi Amugsi[3]

1 School of Population and Development Sciences, University of Kinshasa, Kinshasa, Democratic Republic of The Congo, 2 Faculty of Health, School of Public Health, University of Technology Sydney, Sydney, Australia, 3 African Population and Health Research Center (APHRC), Nairobi, Kenya

☯ These authors contributed equally to this work.
* zacharie.tsala.dimbuene@gmail.com, zacharie.dimbuene@unikin.ac.cd

**Data Availability Statement:** Data are available upon request at https://dhsprogram.com/data/available-datasets.cfm.

**Funding:** The author(s) received no specific funding for this work.

## Abstract

### Background

Intimate partner violence (IPV) has increasingly received attention in the last three decades. However, IPV-related studies in both high- and low- and middle-income countries adopted a victim-oriented perspective in which men are perpetrators and women, the victims. Using socio-cultural and resource theories as guiding frameworks, this paper assessed the associations between men's education and IPV in Central Africa, using nationally representative data of married and cohabiting women of reproductive ages.

### Methods

Data included in the analyses come from Demographic and Health Surveys (DHSs) in the Democratic Republic of the Congo (DRC), Cameroon, Gabon, and Chad. Analyzed sub-samples consisted of 3421, 5023, 3930, and 3221 married/cohabiting women of reproductive ages in Chad, DRC, Cameroon, and Gabon, respectively.

### Results

Findings indicated significant variations of IPV prevalence within and across countries. Previous research demonstrated that men's education is a protective factor in health-related studies. The present study, however, provide no clear evidence on the linkages between men's education and IPV. In contrast, the paper substantiated that highly educated women were at higher risks of IPV when spouses/partners were less educated.

### Conclusion

These findings have policy and programmatic implications because they might impede progress towards SDG goals on the elimination of all forms of violence against girls and women in Central Africa, which recorded the worst development indicators in sub-Saharan Africa.

**Competing interests:** The authors have declared that no competing interests exist.

On a methodological note, studies are increasingly using pooled data to increase statistical power. Those studies can be very limited to devise effective IPV—interventions since they mask geographical variations within and across countries. More effective IPV—interventions should be culturally rooted and accounting for geographical variations because some areas are more affected than others.

## Introduction

Intimate partner violence (IPV), which refers to harmful behaviours such as physical violence, psychological/emotional abuse, sexual coercion, and controlling behaviours committed in intimate relationships remains the most common form of violence against girls and women in the world [1–3]. IPV can be perpetrated by male or female spouses, former partners, and cohabiting partners [4,5]; however, adolescent girls and young women constitute by far the largest share of IPV victims [6,7]. In 2018, evidence suggests that approximately 30% of all women had suffered some form of IPV in their lifetime, and about 10% had experienced IPV in the last 12 months preceding the survey [1]. The largest burden of IPV occurs in low- and middle-income countries (LMICs), especially in sub-Saharan Africa (SSA) [8]. Moreover, previous studies on the geographical variations of IPV found that IPV prevalence is higher in Central Africa compared with other sub-regions in SSA [6,9].

Although men experience IPV; women are at high risks of IPV. Several factors could explain this issue. These include socio-cultural norms that justify wife beating, promote male dominance over female conduct, and women's limited access to wealth [10]. Other factors include unemployment, primary infertility, polygamous unions [11], lower educational attainment, high parity [12], rural residence [13], abuse during childhood, childhood exposure to IPV, social isolation [14], alcohol problems, and drug use [4]. IPV is widely identified as a violation of women's fundamental human rights [10]. Indeed, evidence suggests that IPV increases women's risk for HIV infections, sexual transmitted infections, injuries, depression, induced abortion, premature birth, low birth weight, alcohol use, and homicide-related deaths [1]. Also, IPV hinders women's participation in the socio-economic, with the consequential negative effect on their economic wellbeing as well as their respective countries national growth and development [10].

Evidence on gender differences in IPV perpetration remains largely inconclusive [15,16], and most studies in SSA adopted a victim-oriented perspective labelling men as perpetrators of IPV and women as victims [8]. As such, IPV—related studies in SSA and worldwide increasingly focused on women's socio-demographic characteristics to further our understanding of IPV [17], neglecting the potential influences of men's characteristics that increase (or decrease) women's risks of IPV [8]. For example, partner's lower educational attainment increased women's risks of IPV [7,11,18]. Associations between partner's education and women's risks of IPV are currently understudied, although evidence suggests that men with lower educational levels were more likely to justify IPV [19], and could likely influence their attitudes in perpetrating IPV.

### Men's education and IPV: Theoretical perspective

Previous studies explored several theories to unveil the mechanisms through which men perpetrate IPV towards girls and women. This paper utilizes social—structural theory [20] and resource theory to further our understanding on IPV in Central Africa [21].

## Social—structural theory

This theory postulates that men's perpetration of IPV is a conditioned response to stressors associated with limited resources or inequalities in socioeconomic situations such as education, employment, and wealth [22–25]. Thus, men who are disadvantaged or marginalized did experience "socially structured stress" which increases their likelihood to engage in violent behaviours, often against intimate partners [22,25]. Thus, men could use violence against intimate partners as an adaptation to socially and economic unfavourable conditions [26].

Empirical evidence around the world supports the theory [22,25,27,28]. For instance, previous research showed that prevalence of IPV increased with a decreasing proportion of individuals with high school education, increased proportion of unemployed, and those below the poverty line [22]. Indeed, increased levels of unemployment and poverty associated with low level of education [29,30] could increase economic and social stress among men [31], with a corresponding increase of IPV perpetration [26,27,32].

## Resource theory

Under this theory, power imbalances in key resources (e.g., education, employment, and income) mainly explain why individuals perpetrate IPV against their intimate partners [21]. Proponents of the theory suggest that men who lack adequate material resources to influence or control their partners might resort to violence to re-establish dominance and maintain control in intimate relationships [21,33]. For instance, partners who have low levels of education, are unemployed, and poor but want to maintain control in intimate relationships may resort to violence to assert their power or maintain control [33]. Thus, the lesser resources a man has, the more likely he will engage in violent behaviours in intimate relationships. It is important to point out that having more resources does not necessarily imply lower rates of IPV. Previous research suggests, in some cases, that having more resources does increase men's likelihood to perpetrating IPV, especially when female partners are financially dependent [34]. According to the resource theory, it is possible that an increase in women's resource could affect their traditional gender role performance, which can trigger violence behaviours from male partners [35].

Previous studies which empirically tested the resource theory provide mixed findings [35–38]. Available evidence suggests that men with limited resources such as lower educational attainment are more likely to perpetrate IPV [39,40]. A study conducted in the United States revealed that annual household income was a major predictor of IPV, especially among minority ethnic groups with low education and high unemployment rates [27]. A multi-country study in SSA reported that higher education is a major protective factor against IPV [41]; however, further research is needed to better understand how husband/partner's education affect the magnitude of IPV in Central Africa.

The two theories (social-structural theory and resource theory) stressed the importance of education as a major factor that could influence men's perpetration of IPV. Therefore, the present research is grounded on social-structural and resource theories to further our understanding on the associations between men's educational level and IPV in Central Africa. Considering the importance of education on employment and poverty eradication, the paper contributes to existing literature on education and IPV by focusing on men's education and tests (*a*) the associations between men's educational attainment and IPV; and (*b*) the effects of education gap between spouses/partners and IPV. This analysis is critical because it has the potential to identify pathways for policy and intervention designs to address the high prevalence of IPV in the central Africa sub-region.

## Methods

### Study setting

The study setting is Central Africa. Central Africa offers a unique case to study the effects of partners' education on IPV for two main reasons. First, Central Africa has the worst indicators on IPV as mentioned above. Indeed, previous research reported that IPV prevalence was higher in Central Africa compared with other regions in SSA [6,42]. Second, previous research highlighted the economic costs of IPV worldwide [43,44]. Moreover, studies showed that Central Africa exhibited the worst socio-economic indicators (e.g., corruption index; democratic accountability; law and order) [45]. As such, IPV might be seen as a double burden for socio-economically affected settings.

### Data source

The paper utilizes available data from the Demographic and Health Surveys (DHSs) in selected countries of Central Africa: Cameroon, Chad, the Democratic Republic of the Congo (DRC), and Gabon. Countries were selected based on data availability. DHS datasets are readily available to the public on the DHS website, https://dhsprogram.com/. DHSs are nationally representative surveys, using a two-stage sampling design. The first stage of sampling involved the selection of sample points or clusters from an updated master sampling frame constructed in accordance with country's administrative divisions or domains. These domains were further stratified into urban and rural areas. In the urban areas, neighbourhoods were sampled from cities and towns whereas villages and chiefdoms were sampled for rural areas. The clusters were selected using systematic sampling with probability proportional to size (PPS). Household listing was then conducted in all the selected clusters to provide a complete sampling frame for the second stage selection. The second stage of selection involved the systematic sampling of the households listed in each cluster, and households to be included in the survey were randomly selected. The rationale for the second stage selection was to ensure adequate numbers of completed individual interviews to provide estimates for key indicators with acceptable precision. All men and women aged 15–59 and 15–49, respectively, in the selected households were eligible to participate in the survey if they were either usual residents of the household or visitors present in the household on the night before the survey. After eliminating women with missing information on the "number of cowives" which defines whether a woman lived in polygamous marriages, the final sub-sample consisted of 3421, 5023, 3930, and 3221 in Chad, DRC, Cameroon, and Gabon, respectively. Data are publicly available upon request to DHS Program (https://dhsprogram.com/data/dataset_admin/index.cfm) and were accessed on February 3, 2023.

This paper reports on findings from married and cohabiting women in individual record files to construct the outcome and independent variables. Analyses were restricted to married and cohabiting women because polygynous unions are easier to define in the context of the marriage and cohabitation, and not all women were interviewed for this sensitive module about domestic violence. The DHS collected information on households, women and men of reproductive ages, anthropometric measures, contraception and family planning, among others [46].

### Ethics statement

Ethical approvals were obtained from the national ethics committees in all countries before the survey was conducted. Written informed consent was obtained from every participant before they were allowed to participate in the survey. The DHS Program, USA, granted the authors

permission to use the data. Since the data were completely anonymous, the authors did not seek further ethical clearance for this study.

## Variables measurement

**Outcome.** The present study is interested in intimate partner violence (IPV), including physical, emotional, and sexual violence [47,48]. The sub-components of IPV were derived from the domestic violence module. In this optional module, questions were asked about domestic violence in the last 12 months, based on a modified version of the conflict tactics scale [49,50]. Questions for each sub-component and responses are summarized in Table 1 below.

## Key independent variables

**Education.** Most studies on IPV used "education" as a categorical variable [17,51–53]. Although these studies provided insights about the association between education and IPV, they might be limited in terms of strategic and programmatic implications to eradicate IPV. DHSs offer the opportunity to capture educational levels in years completed. This paper adopts this operationalization since treating educational clusters as homogenous groups can mask within-cluster inequalities. This is a derived variable, at country level, from the variables *v106* (Educational level) and *v107* (Grade at the level). There are slight variations in the countries under study; however, overall, the educational levels ranged from 0 to 20. Invalid cases were excluded from the analyses.

**Difference in education level between spouses/partners.** Previous studies also tested the association between education gap between spouses and IPV [53] and found that the risks of IPV increased when the wife/female partner is more educated than husband/male partner. In this study, education gap between spouses/partners was computed as the difference between the husband/male partner and wife/female partner in educational levels (in years completed). Conceptually, the difference is positive if husband/male partner is more educated than wife/female partner, and negative if otherwise.

**Polygamous unions.** Married/cohabiting women were asked to report the number of other wives that the husband had [47,54,55]. Women who indicated that their husbands/

**Table 1. Questions and responses about intimate partner violence.**

| Components of IPV and questions | Responses | Operational definition |
|---|---|---|
| **Physical violence (7 items)** | | |
| 1. Husband ever pushed, shook, or threw something at her<br>2. Husband slapped her<br>3. Husband punched her with his fist or something harmful<br>4. Husband kicked or dragged her<br>5. Husband strangled or burnt her<br>6. Husband threatened her with a knife, gun, or other weapons<br>7. Husband twisted her arm or pulled her hair | Responses included 0 "Never"; 1 "Often"; 2 "Sometimes"; and 3 "Yes, but not in the last 12 months" | The item was recorded 0 "No" if wife reported "Never" or "Yes, but not in the last 12 months" and 1 if wife reported "Often" or "Sometimes". The new variable ranged from 0 to 7 |
| **Emotional violence (3 items)** | | |
| 1. Husband humiliated her<br>2. Husband threatened to harm her<br>3. Husband insulted or made her feel bad | Responses included 0 "Never"; 1 "Often"; 2 "Sometimes"; and 3 "Yes, but not in the last 12 months" | The item was recorded 0 "No" if wife reported "Never" or "Yes, but not in the last 12 months" and 1 if wife reported "Often" or "Sometimes". The new variable ranged from 0 to 3 |
| **Sexual violence (3 items)** | | |
| 1. Husband ever physically forced wife into unwanted sex<br>2. Husband ever forced wife into other unwanted sexual acts<br>3. Respondent has been physically forced to perform sexual acts she didn't want to | Responses included 0 "Never"; 1 "Often"; 2 "Sometimes"; and 3 "Yes, but not in the last 12 months" | The item was recorded 0 "No" if wife reported "Never" or "Yes, but not in the last 12 months" and 1 if wife reported "Often" or "Sometimes". The new variable ranged from 0 to 3 |

partners had no other wives lived in monogamous marriages. In contrast, those who indicated that their husbands/partners had at least one or more other wives lived in polygamous unions. Therefore, the variable 'polygyny' is a dichotomous variable taking the value "1" if the woman lived in polygamous unions and "0" otherwise.

**Urban residence.**  This is a binary variable coded "1" for urban women and "0" for rural women.

Previous research suggested urban advantage in health-related studies [56]. This paper posits that since education is usually higher in urban areas compared with rural areas, the urban advantage is still justifiable in IPV-related studies. Some studies showed that IPV was more prevalent in rural settings compared to urban [17,57].

**Control variables.**  Based on previous research, control variables were included: Household wealth index (HWI), media exposure, and attitudes towards domestic violence. HWI in the original dataset was categorized as 'poorest', 'poorer', 'middle', "richer", and 'richest'. This variable was recorded into three categories: Poor (bottom 40%) coded "1"; Middle (20%) coded 2 and Rich (top 40%) coded "3". The detailed discussion on the construction of HWI has been published elsewhere [58,59]. The index of media exposure was created from three variables: the frequency of watching television, listening to radio, or reading newspapers/magazines. Responses to these variables were '0' if respondent reported 'not at all', '1' for 'less than once a week', and '2' for 'at least once a week'. Responses were recorded into '0 = No' for 'not at all' and '1 = Yes' for 'less than once a week' and 'at least once a week'. Finally, a dichotomous variable was created from a composite of exposure to the three media sources and defined as "0 = No" for married women who scored '0' on the three items and '1 = Yes' if women's score was higher or equal to '1'. Finally, attitudes towards violence correlated with IPV [60–64]. The variable 'justification of violence' was derived from questions asking married/cohabiting women if it is justified for a husband to beat his wife for the following reasons: (*i*) burning food, (*ii*) arguing with him, (*iii*) going out without telling him, (*iv*) neglecting the children, and (*v*) refusing to have sexual intercourse with him. A binary variable was created from these five reasons to reflect the attitudes towards wife beating. Justification of violence was therefore coded as '0 = No' if a woman disagreed with the five reasons and '1 = Yes' if she agreed to at least one of these reasons.

### Analytical strategy

**Descriptive analyses.**  Most studies on IPV analyzed data on domestic violence either at national level or pooled data to increase statistical power. In this paper, the outcome (IPV) was spatially analyzed to unveil the heterogeneity of IPV across provinces in each country. Studies found that the likelihood of IPV was higher among women in polygynous unions [47,60,65–67].

### Modelling strategy

The methods of data analysis include an examination of the association between the key independent variables and IPV (bivariate analyses) for the selected key independent variables, using unadjusted odd ratios (OR) derived from logistic regression. Additionally, these models were extended to include interactions between husband/partner's education and (*i*) polygyny; (*ii*) urban residence; (*iii*) women's education. Another model was estimated to include the education gap between spouses/partners and tested interactions between education gap and polygyny (Model 5). Statistical analyses were performed using STATA version 18 *SE*.

**Preliminary analyses.**  Before fitting multivariate models, assumptions about logistic regression were checked carefully. In particular, multicollinearity tests and statistical significance of the associations between the outcome and independent variables were examined.

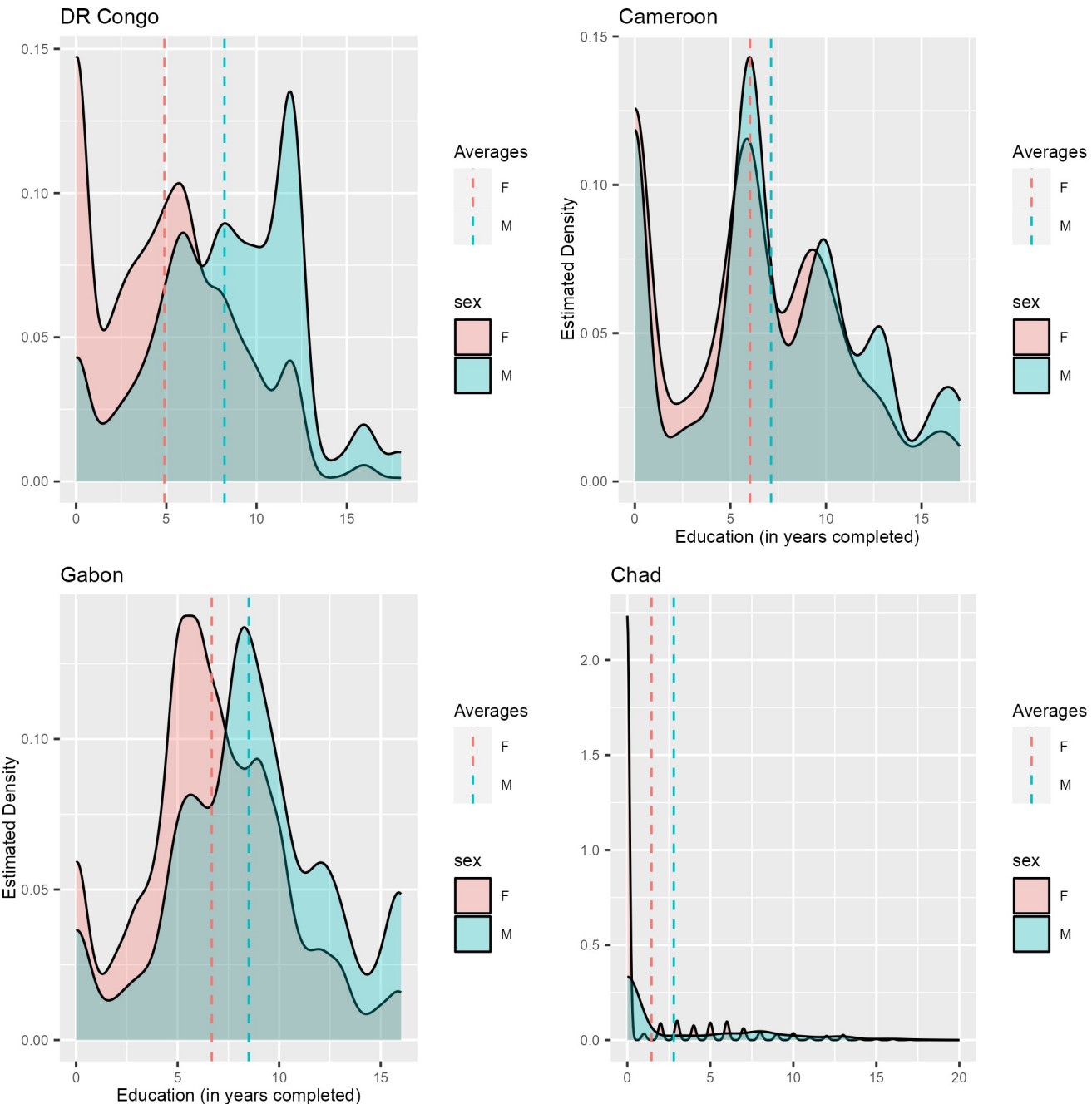

**Fig 1. Density plots of education among women and men in Central Africa.** Dashed lines indicate average education level. Source: DHSs datasets in selected countries.

Using variance tolerance, known as Variance Inflation Factor (VIF), the tests revealed no problem of multicollinearity. All VIF values were less than 2.

**Goodness-of-fit of the models and the influence of the outliers.** Another issue discussed in multivariate logistic regression is the extent to which estimated models significantly fit the data. The tests included log-likelihood, test of Hosmer-Lemeshow, Pearson's Chi-squared of the model and the Receiver Operating Characteristic (ROC). The influence of outliers on the

**Table 2. Prevalence of (1) physical violence; (2) emotional violence; (3) sexual violence; and (4) intimate partner violence in Central Africa.**

| A. Democratic Republic of the Congo (*N = 5,023*) | | | | |
|---|---|---|---|---|
| Province/Region | (1) | (2) | (3) | (4) |
| Kinshasa | 21.0 | 20.8 | 10.7 | 33.8 |
| Bandundu | 34.0 | 27.2 | 25.5 | 47.7 |
| Kongo Central | 27.5 | 25.8 | 11.5 | 37.3 |
| Equateur | 36.0 | 29.6 | 18.0 | 46.0 |
| Kasai Occidental | 42.5 | 36.9 | 29.2 | 57.6 |
| Kasai Oriental | 36.4 | 38.7 | 24.7 | 53.3 |
| Katanga | 24.9 | 25.2 | 18.5 | 40.8 |
| Maniema | 29.7 | 27.9 | 19.6 | 40.2 |
| North Kivu | 13.4 | 24.7 | 17.6 | 36.5 |
| Orientale | 26.2 | 23.0 | 12.3 | 35.8 |
| South Kivu | 31.7 | 39.8 | 19.6 | 49.9 |
| Total | 29.4 | 29.1 | 18.8 | 43.5 |
| **B. Cameroon (*N = 3,930*)** | | | | |
| Adamawa | 7.2 | 11.3 | 3.3 | 15.8 |
| Centre | 28.8 | 26.2 | 11.5 | 38.8 |
| Littoral | 15.7 | 18.9 | 8.6 | 29.1 |
| East | 23.1 | 26.5 | 6.1 | 36.7 |
| Far-North | 10.8 | 11.8 | 2.6 | 17.6 |
| North | 24.8 | 19.6 | 4.5 | 30.1 |
| North-West | 17.5 | 31.8 | 6.8 | 37.7 |
| West | 14.3 | 34.9 | 5.6 | 40.5 |
| South | 25.1 | 26.3 | 8.0 | 35.9 |
| South-West | 10.0 | 22.6 | 4.6 | 25.3 |
| Total | 17.7 | 23.0 | 6.2 | 30.8 |
| **C. Gabon (*N = 3,221*)** | | | | |
| Estuaire | 26.7 | 24.2 | 10.6 | 38.1 |
| Haut-Ogooue | 21.9 | 15.2 | 3.8 | 28.0 |
| Moyen-Ogooue | 27.9 | 25.5 | 11.7 | 37.3 |
| Ngounie | 33.2 | 35.8 | 18.4 | 48.0 |
| Nyanga | 23.2 | 30.4 | 16.5 | 40.7 |
| Ogooue Maritime | 30.4 | 19.3 | 2.7 | 34.4 |
| Ogooue-Ivindo | 44.3 | 32.2 | 13.5 | 51.5 |
| Ogooue-Lolo | 34.2 | 27.1 | 9.1 | 49.0 |
| Woleu-Ntem | 26.2 | 27.6 | 7.4 | 36.1 |
| Total | 29.8 | 26.4 | 10.4 | 40.3 |
| **D. Chad (*N = 3,421*)** | | | | |
| Batha | 11.9 | 10.8 | 9.5 | 19.3 |
| Borkou | 6.9 | 8.0 | 3.7 | 14.9 |
| Tibesti | 6.9 | 8.0 | 3.7 | 14.9 |
| Chari Baguirmi | 8.0 | 9.6 | 8.9 | 16.3 |
| Guera | 5.6 | 10.9 | 0.7 | 12.1 |
| Hadjer-Lamis | 4.9 | 6.5 | 5.5 | 13.5 |
| Kanem | 5.2 | 5.7 | 11.9 | 14.4 |
| Lac | 5.9 | 3.4 | 8.4 | 10.1 |
| Logone Occidental | 18.3 | 18.4 | 2.1 | 22.2 |

*(Continued)*

**Table 2.** (Continued)

| Logone Oriental | 28.6 | 25.9 | 4.7 | 40.0 |
| Mandoul | 14.6 | 16.3 | 5.5 | 23.3 |
| Mayo Kebbi East | 23.3 | 21.6 | 7.3 | 30.6 |
| Mayo Kebbi West | 32.6 | 23.4 | 22.6 | 44.8 |
| Moyen chari | 9.4 | 10.3 | 0.0 | 16.1 |
| Ouaddae | 7.4 | 8.4 | 0.8 | 12.8 |
| Salamat | 5.5 | 7.7 | 3.6 | 11.6 |
| Tandjile | 27.7 | 27.4 | 11.9 | 39.4 |
| Wadi Fira | 13.5 | 17.8 | 9.3 | 19.5 |
| N'djamena | 12.3 | 14.3 | 6.7 | 18.2 |
| Barh el Gazal | 9.9 | 11.5 | 13.1 | 17.2 |
| Ennedi East | 3.9 | 0.9 | 2.0 | 5.0 |
| Ennedi West | 3.9 | 0.9 | 2.0 | 5.0 |
| Sila | 9.1 | 10.5 | 8.7 | 13.6 |
| Total | 12.0 | 12.1 | 6.6 | 18.9 |

estimates was examined using a plot of the residuals and predicted probabilities of the outcome to check for covariate patterns and overdispersion. Residuals with absolute values more than 1 indicate a problematic covariate pattern that can undermine the goodness-of-fit of the models. However, the plots depicted no residual values above 1 or overdispersion issues.

## Findings

### Descriptive results

**Education among married/cohabiting women and husbands/partners in Central Africa.** Fig 1 presents the distribution of number of years of education completed at the time

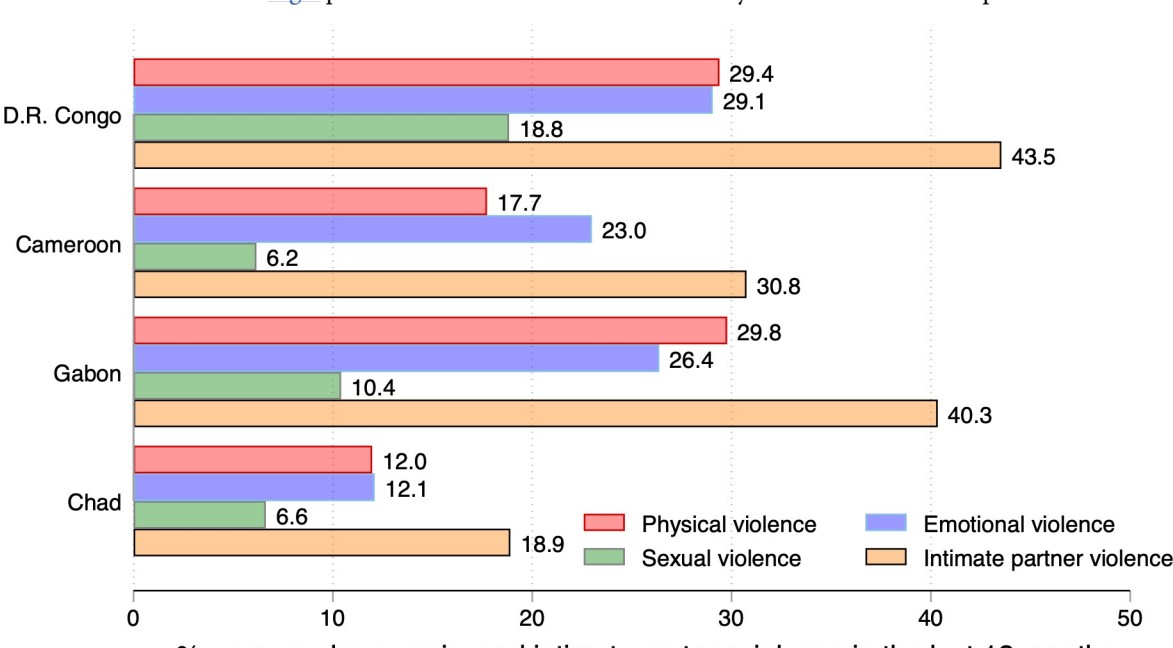

**Fig 2. Prevalence of physical, emotional, sexual violence, and intimate partner violence among married/cohabiting women in Central Africa.** Source: DHSs datasets in selected countries.

**Table 3. Estimated odd ratios (OR) of husband's education on intimate partner violence in the Democratic Republic of the Congo.**

| Variables | Model 1 | Model 2 | Model 3 | Model 4 | Model 5 |
|---|---|---|---|---|---|
| **Main effects** | | | | | |
| Women's education (in single years) | 0.998 | 0.998 | 1.001 | 1.078*** | |
| | (0.974–1.022) | (0.974–1.022) | (0.977–1.026) | (1.023–1.136) | |
| Husband/partner's education | 0.993 | 0.993 | 1.013 | 1.029* | |
| | (0.968–1.019) | (0.964–1.023) | (0.985–1.042) | (0.996–1.063) | |
| Polygynous unions (Ref.: Monogamous) | 1.508*** | 1.480* | 1.509*** | 1.508*** | 1.475** |
| | (1.197–1.900) | (0.960–2.281) | (1.196–1.905) | (1.193–1.905) | (1.077–2.021) |
| Urban residence (Ref.: Rural) | 0.932 | 0.933 | 2.026** | 0.961 | 0.915 |
| | (0.659–1.319) | (0.660–1.318) | (1.158–3.544) | (0.676–1.368) | (0.652–1.284) |
| Education difference (in single years) | | | | | 0.995 |
| | | | | | (0.970–1.021) |
| **Interaction effects** | | | | | |
| Husband's education X Polygyny | | 1.002 | | | |
| | | (0.951–1.057) | | | |
| Husband's education X Urban residence | | | 0.923*** | | |
| | | | (0.878–0.972) | | |
| Husband's education X Women's education | | | | 0.992*** | |
| | | | | (0.988–0.997) | |
| Education difference X Polygyny | | | | | 1.007 |
| | | | | | (0.945–1.074) |
| Observations | 5,023 | 5,023 | 5,023 | 5,023 | 5,023 |

95% confidence intervals in parentheses.

Statistical significance

*** p<0.01

** p<0.05

* p<0.1.

Models 1–5 controls for household wealth index, exposure to media, and attitude towards violence.

of the surveys in the selected countries. Different patterns were observed across countries. While in Chad, the distribution of education (in completed years) was highly skewed on the left, with women and men falling mostly at the very bottom, it was more equilibrated in Gabon despite men's advantage in all other countries. On average, men completed 8.2 years, 7.1 years, 8.5 years and 2.8 years in the DRC, Cameroon, Gabon, and Chad, respectively. The corresponding figures for women were 4.9 years, 6.0 years, 6.7, and 1.5 year in the DRC, Cameroon, Gabon, and Chad, respectively.

**Prevalence of intimate partner violence in Central Africa.** Table 2 (cols. 1–4) and Fig 2 present the prevalence of IPV components (cols. 2–4) and the estimated prevalence of IPV among married/cohabiting women in Central Africa.

In the DRC, Gabon, and Chad, IPV and its components showed some patterns, with physical violence being the most prevalent, followed by emotional violence and sexual violence. Among the three countries, physical violence was higher in the DRC (29.4%) and Gabon (29.8%) than in Chad (12.0%). Likewise, emotional violence among married/cohabiting women was higher in the DRC (29.1%) and Gabon (26.4%) than in Chad (12.1%). Sexual violence recorded the lowest prevalence in the three countries, ranging between 6.6% in Chad to 18.8% in the DRC. In Cameroon, findings indicated that emotional violence was more

**Table 4. Estimated odd ratios (OR) of husband's education on intimate partner violence in Cameroon.**

| Variables | Model 1 | Model 2 | Model 3 | Model 4 | Model 5 |
|---|---|---|---|---|---|
| **Main effects** | | | | | |
| Women's education (in single years) | 1.014 | 1.015 | 1.017 | 1.123*** | |
| | (0.985–1.045) | (0.984–1.046) | (0.987–1.048) | (1.067–1.182) | |
| Husband/Partner's education (in single years) | 1.027** | 1.025* | 1.064*** | 1.108*** | |
| | (1.000–1.054) | (0.996–1.055) | (1.027–1.101) | (1.066–1.151) | |
| Polygynous unions (Ref.: Monogamous) | 1.022 | 0.978 | 1.047 | 1.096 | 0.981 |
| | (0.781–1.338) | (0.617–1.552) | (0.798–1.374) | (0.831–1.447) | (0.737–1.304) |
| Urban residence (Ref.: Rural) | 0.709** | 0.709** | 1.133 | 0.723** | 0.722** |
| | (0.535–0.939) | (0.535–0.939) | (0.741–1.732) | (0.555–0.943) | (0.544–0.959) |
| Education difference (in single years) | | | | | 1.018 |
| | | | | | (0.992–1.045) |
| **Interaction effects** | | | | | |
| Husband's education X Polygyny | | 1.010 | | | |
| | | (0.950–1.074) | | | |
| Husband's education X Urban residence | | | 0.935*** | | |
| | | | (0.896–0.977) | | |
| Husband's education X Women's education | | | | 0.988*** | |
| | | | | (0.983–0.992) | |
| Education difference X Polygyny | | | | | 0.963 |
| | | | | | (0.903–1.028) |
| Observations | 3,930 | 3,930 | 3,930 | 3,930 | 3,930 |

95% confidence intervals in parentheses.

Statistical significance

*** p<0.01

** p<0.05

* p<0.1.

Models 1–5 controls for household wealth index, exposure to media, and attitude towards violence.

prevalent (23.0%), than physical violence (17.7%) and sexual violence (6.6%). Results showed that 18.9% of married/cohabiting women experienced IPV in the last 12 months. The corresponding figures for the DRC, Cameroon, and Gabon were 43.5%, 30.8%, and 40.3%, respectively.

## Multivariate results

Tables 3–6 present adjusted odd ratios (AOR) between men's education (Model 1), controlling for women's education, polygyny and urban residence, and other factors (household wealth index, exposure to media, ad attitudes towards violence). Findings in Model 1 presented mixed results across countries. Men's education had a negative association with IPV (DRC and Gabon); however, the associations did not reach statistical significance. In contrast, men's education was positively and significantly associated with IPV in Cameroon and Chad. An additional year of education among husbands/partners increased the odds of IPV by 2.7% and 3.2% in Cameroon and Chad, respectively. Model 1 (Table 3) showed that polygyny was positively and significantly associated with IPV in the DRC. Living in polygamous marriages increased the likelihood of experiencing IPV in last 12 months by 51%. In Cameroon, married/cohabiting women in urban areas were significantly less likely to experience IPV in the

**Table 5. Estimated odd ratios (OR) of husband's education on intimate partner violence in Gabon.**

| Variables | Model 1 | Model 2 | Model 3 | Model 4 | Model 5 |
|---|---|---|---|---|---|
| **Main effects** | | | | | |
| Women's education (in single years) | 1.024 | 1.024 | 1.024 | 1.104** | |
| | (0.986–1.063) | (0.986–1.065) | (0.986–1.063) | (1.000–1.219) | |
| Husband/partner's education (in single years) | 0.993 | 0.980 | 0.985 | 1.047 | |
| | (0.957–1.030) | (0.945–1.015) | (0.945–1.027) | (0.985–1.114) | |
| In polygynous union = 1, YES | 0.962 | 0.486* | 0.963 | 0.994 | 0.926 |
| | (0.678–1.366) | (0.231–1.023) | (0.678–1.367) | (0.694–1.425) | (0.646–1.327) |
| Urban residence = 1, Urban | 1.181 | 1.189 | 1.101 | 1.186 | 1.186 |
| | (0.903–1.545) | (0.910–1.555) | (0.667–1.819) | (0.903–1.557) | (0.907–1.552) |
| Education difference (in single years) | | | | | 0.984 |
| | | | | | (0.951–1.018) |
| **Interaction effects** | | | | | |
| Husband's education X Polygyny | | 1.080* | | | |
| | | (0.995–1.172) | | | |
| Husband's education X Urban residence | | | 1.009 | | |
| | | | (0.959–1.061) | | |
| Husband's education X Women's education | | | | 0.992* | |
| | | | | (0.984–1.000) | |
| Education difference X Polygyny | | | | | 1.019 |
| | | | | | (0.916–1.135) |
| Observations | 3,221 | 3,221 | 3,221 | 3,221 | 3,221 |

95% confidence intervals in parentheses.

Statistical significance

*** p<0.01

** p<0.05

* p<0.1.

Models 1–5 controls for household wealth index, exposure to media, and attitude towards violence.

last 12 months. Living in urban settings in Cameroon decreased the likelihood to experience IPV by 30%.

Models 2–4 (Tables 3–6) tested interactions between men's education and (*a*) polygyny; (*b*) urban residence; and (*c*) women's education. Results from multiplicative models varied across countries. Regarding the interactions between men's education and polygyny, results did not reach statistical significance in the DRC and Cameroon. The interaction term was marginally significant in Gabon, and statistically significant in Chad. The interaction between men's education and urban residence showed inverse relationships in DRC, Cameroon, and Chad. Finally, the interaction term between men's education and women's education was statistically significant in all the four countries. Multiplicative models can be poorly understandable without visualization which provides a more intuitive ways to capture the sense of interactions between the variables of interest. Therefore, interactions are plotted in Figs 3–5.

First, there is evidence that the likelihood of IPV increased as husbands/partners' education increased for women living in polygamous marriages in Cameroon, Gabon, and Chad but not in the DRC (Fig 3). Second, findings evidenced that as husbands/partners' education increased, the likelihood of IPV decreased in urban areas in the DRC and Gabon. Third, Fig 3

**Table 6. Estimated odd ratios (OR) of husband's education on intimate partner violence in Chad.**

| Variables | Model 1 | Model 2 | Model 3 | Model 4 | Model 5 |
|---|---|---|---|---|---|
| **Main effects** | | | | | |
| Women's education (in single years) | 1.014 | 1.014 | 1.024 | 1.088*** | |
| | (0.970–1.059) | (0.970–1.060) | (0.978–1.071) | (1.022–1.158) | |
| Husband/Partner's education (in single years) | 1.032** | 1.014 | 1.054*** | 1.054*** | |
| | (1.001–1.064) | (0.981–1.050) | (1.018–1.090) | (1.022–1.088) | |
| Polygynous unions (Ref.: Monogamous) | 1.153 | 0.940 | 1.142 | 1.158 | 1.045 |
| | (0.912–1.458) | (0.710–1.246) | (0.903–1.444) | (0.914–1.467) | (0.810–1.349) |
| Urban residence (Ref.: Rural) | 0.987 | 1.015 | 1.448 | 1.013 | 1.057 |
| | (0.646–1.507) | (0.665–1.550) | (0.886–2.367) | (0.665–1.544) | (0.697–1.604) |
| Education difference (in single years) | | | | | 1.013 |
| | | | | | (0.979–1.048) |
| **Interaction effects** | | | | | |
| Husband's education X Polygyny | | 1.061*** | | | |
| | | (1.015–1.109) | | | |
| Husband's education X Urban residence | | | 0.922*** | | |
| | | | (0.875–0.971) | | |
| Husband's education X Women's education | | | | 0.991*** | |
| | | | | (0.986–0.997) | |
| Education difference X Polygyny | | | | | 1.042 |
| | | | | | (0.980–1.109) |
| Observations | 3,427 | 3,427 | 3,427 | 3,427 | 3,427 |

95% confidence intervals in parentheses.

Statistical significance

*** p<0.01

** p<0.05

* p<0.1.

Models 1–5 controls for household wealth index, exposure to media, and attitude towards violence.

showed women's education played a key in understand the likelihood of IPV. When women are more educated (compared with husbands/partners), the prevalence of IPV was higher. Fourth, Fig 4 illustrated the well-known "urban advantage" in health-related studies even though the patterns did vary across countries. For instance, urban advantage was evident with the increase of partner's education. Fifth, Fig 5 highlighted the higher risks of IPV when women were more educated than husbands/partners; and this was consistent across countries.

The study was also interested in estimating the association between the differences of men's education and women's education and IPV in Central Africa (Model 5, Tables 3–6). Fig 6 about spouses' education differences showed interesting features to better capture the associations between education and IPV. The four countries are quite different. In Cameroon and Chad, the highest percentage is located at zero meaning that the differences in education among spouses are quite small. That is not the case in the DRC and Gabon showing that men are more educated than women. However, Model 5 in Tables 3–6 did show no indication of a significant association between spouses' education differences and IPV in Central Africa.

## Discussion

Grounded in socio-cultural and resources theories, and against the victim-oriented perspective mostly used in previous research, this paper investigated specific roles of men's education on

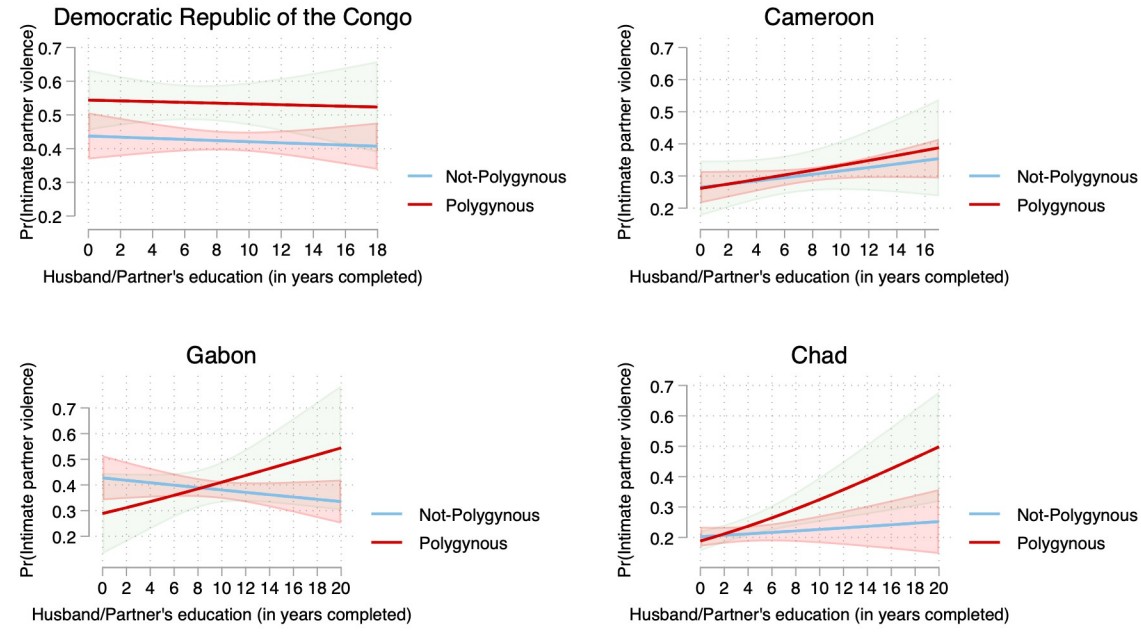

Note: Group difference (Monogamous vs Polygynous unions) is significant ($p < 0.05$) when lines are solid

**Fig 3. Probability of intimate partner violence among married/cohabiting women in Central Africa—interaction between men's education and polygyny.**

IPV among married/cohabiting women in the DRC, Cameroon, Gabon, and Chad. Previous studies have examined the associations between men's education and IPV; however, they were conducted in specific settings [52,68,69]. This study extended existing literature on the inter-linkages between men's education and IPV in various contexts in Central Africa, while exploring the joint effects of men's education and (*a*) polygamous marriages; (*b*) women's education; and (*c*) urban residence. Finally, the paper examined the associations between spouses' education differences and IPV and its interactions with polygamous marriages. In doing so, the paper reinforced the needs of context-specific knowledge for more effective interventions to eradicate IPV in Central Africa and worldwide.

Findings indicated wide variations of IPV prevalence in Central Africa, ranging from 18.9% in Chad to 43.5% in the DRC. Compared with previous studies [1,70,71], findings indicated that IPV prevalence was higher in the DRC, Gabon and Cameroon. Worldwide, it is estimated that one-third of women has suffered from IPV in their lifetime. Findings provide limited support of both sociocultural theory and resource theory. According to sociocultural theory, it was expected that educated men are more open to human rights and therefore, they are less likely to perpetrate violence in intimate relationships. Although an additional year of education among husbands/partners decreased the rates of IPV in the DRC and Gabon, the associations did not reach statistical significance. In contrast, an additional year of men's education in Cameroon and Chad increased the risks of IPV. As mentioned above, most studies are victims-oriented and they examined association between female education and IPV [8,30,68,72]. This paper focused on men's education, and it found mix effects. The fact that male education

## Probability of intimate partner violence
### Interaction effect: Husband/Partner's education and urban residence

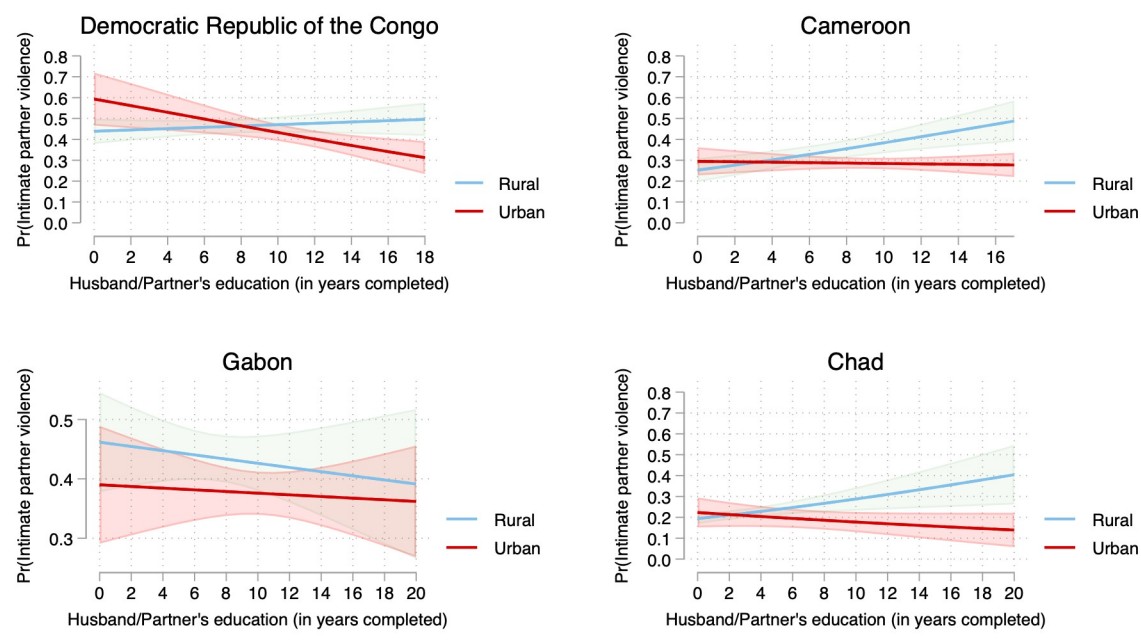

Note: Group difference (Rural vs Urban residence) is significant (*p* < 0.05) when lines are solid

**Fig 4. Probability of intimate partner violence among married/cohabiting women in Central Africa—interaction between men's education and urban residence.**

increased the risks of IPV in Cameroon and Chad calls for a closer examination. It is possible that highly educated men have better jobs and therefore they have more money, which exposes them to more women. This might increase jealousy among wives and/or partners, thereby increasing the levels of IPV.

Findings provided partial support of resource theory. The multiplicative models showed that as men's education increases, the likelihood of IPV increased in Cameroon, Gabon, and Chad, especially among women in polygamous marriages. Findings in Gabon provided an interesting scenario because the risks of IPV decreased among women in monogamous marriages as husbands/partners' education increased. In Cameroon and Chad, even though the risks of IPV increased as men's education increased, the risks of IPV were lower for monogamous marriages compared with polygamous marriages. Additionally, the interactions between men's and women's education provided more convincing support of resource theory. Indeed, when women are more educated than men, the risks of IPV were higher. This is an indication that education gaps might lead to tension between spouses. In a previous study, using pooled data in sub-Saharan Africa, researchers demonstrated that resources do not protect women against abuse [39]. Nonetheless, resource inequality within household was associated with higher risks of abuse. Education among women could increase access to resources through employment, more control over financial resources, more exposure to media and information about women's human rights; and therefore, equipping women to resist against abuse (e.g., IPV) but at the same time, increasing tension in the households which can lead to higher risks of IPV.

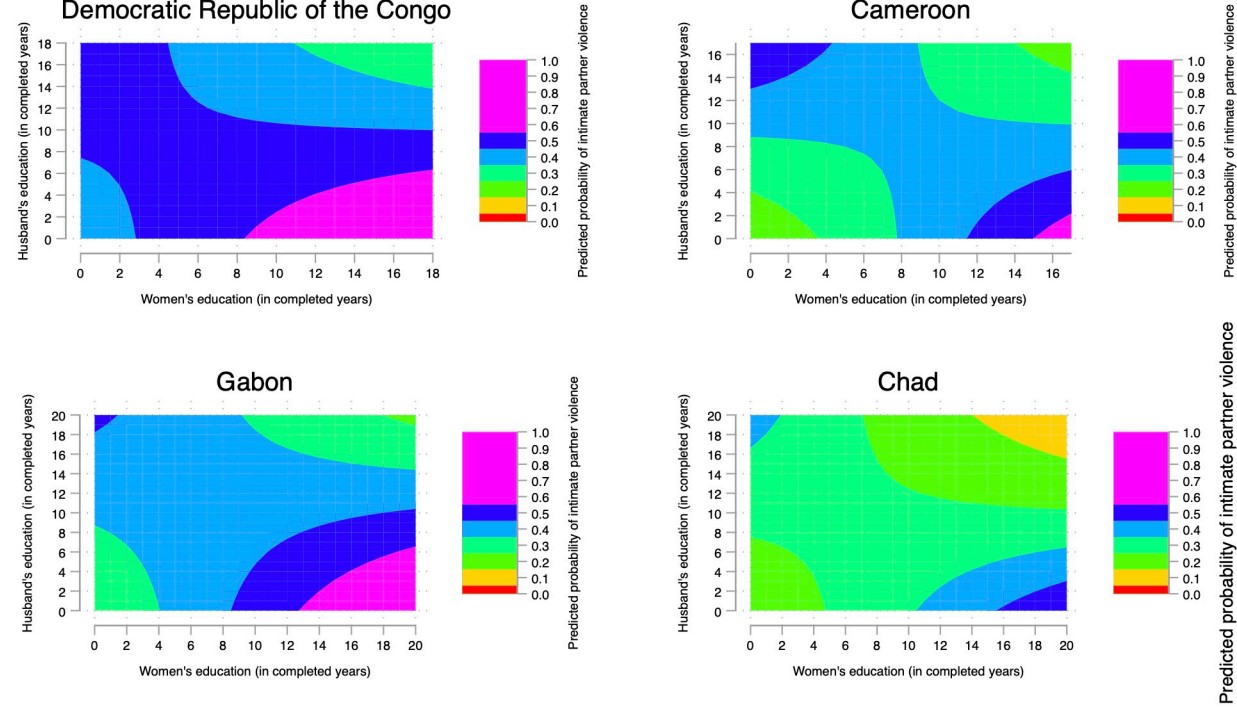

**Fig 5. Probability of intimate partner violence among married/cohabiting women in Central Africa—interaction between men's education and women's education.**

## Study limitations and future research

The paper used recent nationally representative datasets of married/cohabiting women of reproductive ages in four countries in Central Africa (Democratic Republic of the Congo, Cameroon, Gabon, and Gabon). As such, findings from the present study are robust. However, it has some limitations. One of the limitations of the study relies on the cross-sectional nature of the data. Cross-sectional data can only detect associations between the outcome of interest and men's education, specifically. Therefore, no definite conclusion can be drawn regarding the causality between men's education and IPV in Central Africa. Finally, questions asked to women in "Domestic Violence" module referred to a 12-month period. Responses might suffer from recall bias.

Findings showed that women are at higher risks of IPV when they are more educated than their husbands/partners. Future research might be interested in emotional and psychological feelings of less educated men to better understand why they are more likely to engage in IPV behaviours, with a special attention to cultural norms in study settings.

## Conclusion

Men's education has been found as an important protective factor in social- and health-related studies. The present research extended previous studies in adopting a different approach with

## Spouse's education difference

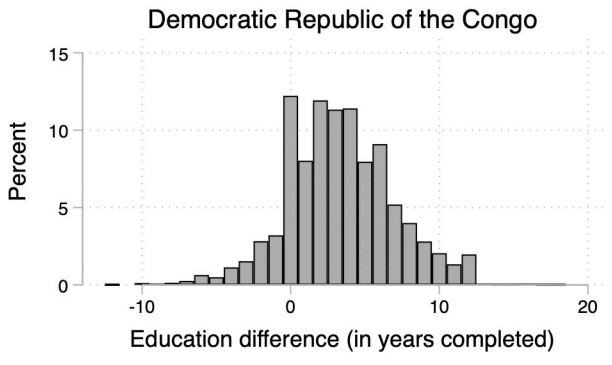
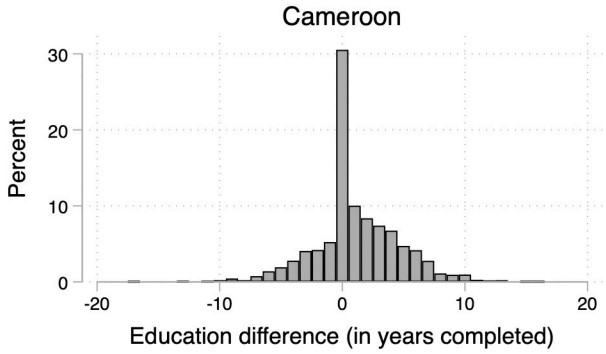
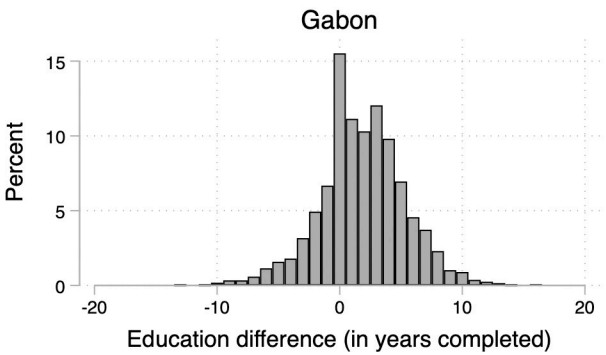
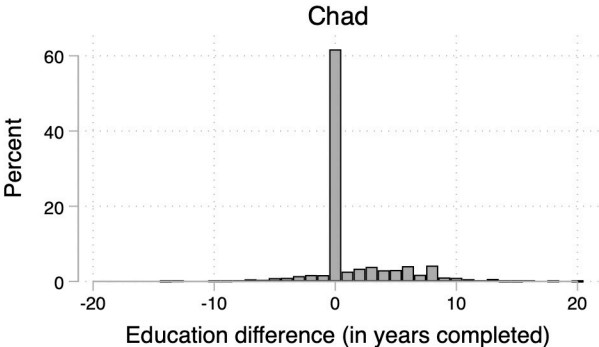

**Fig 6. Differences in spouses' education in Central Africa.**

a focus on men's education as a contributor to IPV in Central Africa. However, findings showed no clear evidence on the association between men's education and IPV. In contrast, findings indicated that highly educated women were at higher risks of IPV when spouses/partners were less educated. These findings have policy and programmatic implications because lower levels of educational among husbands/partners might impede progress towards SDG goals on the elimination of all forms of violence against girls and women in Central Africa. Policies to tackle under-education among men are of chief importance to ensure that human capital between spouses/partners are equalized, as a promising way to reduce IPV among highly educated women and girls in Central Africa.

## Acknowledgments

The authors wish to express their gratitude to the DHS Program, USA, for its generosity in providing them full access to the data. They also wish to acknowledge institutions of the Democratic Republic of the Congo, Cameroon, Gabon, and Chad that played critical roles in the data collection process. The authors also express their gratitude to Ms. Stella Kasura who generously reviewed previous versions of the manuscript.

## Author Contributions

**Conceptualization:** Zacharie Tsala Dimbuene, Bright Opoku Ahinkorah, Dickson Abanimi Amugsi.

**Data curation:** Zacharie Tsala Dimbuene.

**Formal analysis:** Zacharie Tsala Dimbuene, Bright Opoku Ahinkorah, Dickson Abanimi Amugsi.

**Investigation:** Bright Opoku Ahinkorah, Dickson Abanimi Amugsi.

**Methodology:** Zacharie Tsala Dimbuene, Bright Opoku Ahinkorah, Dickson Abanimi Amugsi.

**Project administration:** Zacharie Tsala Dimbuene.

**Software:** Zacharie Tsala Dimbuene.

**Supervision:** Zacharie Tsala Dimbuene, Bright Opoku Ahinkorah, Dickson Abanimi Amugsi.

**Validation:** Zacharie Tsala Dimbuene, Bright Opoku Ahinkorah, Dickson Abanimi Amugsi.

**Visualization:** Zacharie Tsala Dimbuene, Bright Opoku Ahinkorah, Dickson Abanimi Amugsi.

**Writing – original draft:** Zacharie Tsala Dimbuene, Bright Opoku Ahinkorah, Dickson Abanimi Amugsi.

**Writing – review & editing:** Zacharie Tsala Dimbuene, Bright Opoku Ahinkorah, Dickson Abanimi Amugsi.

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
