## [Decision Letter · Decision Letter 0]

26 Feb 2024

PONE-D-23-33735Men’s education and intimate partner violence—Beyond the victim-oriented perspective: Evidence from Demographic and Health Surveys in Central AfricaPLOS ONE

Dear Dr. Tsala Dimbuene,

Thank you for submitting your manuscript to PLOS ONE. After careful consideration, we feel that it has merit but does not fully meet PLOS ONE’s publication criteria as it currently stands. Therefore, we invite you to submit a revised version of the manuscript that addresses the points raised during the review process.

We look forward to receiving your revised manuscript.

Kind regards,

Amos Buh, BSc., MPH, PhD

Academic Editor

PLOS ONE

Journal Requirements:

Reviewers' comments:

Reviewer's Responses to Questions

**Comments to the Author**

1. Is the manuscript technically sound, and do the data support the conclusions?

Reviewer #1: Yes

Reviewer #2: Yes

2. Has the statistical analysis been performed appropriately and rigorously? 

Reviewer #1: Yes

Reviewer #2: Yes

3. Have the authors made all data underlying the findings in their manuscript fully available?

Reviewer #1: Yes

Reviewer #2: Yes

4. Is the manuscript presented in an intelligible fashion and written in standard English?

Reviewer #1: Yes

Reviewer #2: Yes

5. Review Comments to the Author

Reviewer #1: The manuscript should include a section on limitations and, perhaps, directions for future research. The limitations section should be detailed and thorough.

I suggest moving most of the section's “analytical strategy” content to an appendix while keeping the descriptive analysis paragraph and including a brief description and reference to the appendix.

Reviewer #2: - Some sentences in the introduction need clarification

- The manuscript needs to enrich the theoretical framework

-Show the limitations of the study better

- Study procedures are good

- References are up to date

6. PLOS authors have the option to publish the peer review history of their article (what does this mean?). If published, this will include your full peer review and any attached files.

Reviewer #1: **Yes: **Dr. Ali M. AL-Asadi

Reviewer #2: **Yes: **Mohammad Khair Alsalamat

---

## [Author Response · Author response to Decision Letter 0]

19 Mar 2024

See point-by-point responses to reviewers

---

## [Decision Letter · Decision Letter 1]

1 Apr 2024

PONE-D-23-33735R1Men’s education and intimate partner violence—Beyond the victim-oriented perspective: Evidence from Demographic and Health Surveys in Central AfricaPLOS ONE

Dear Dr. Tsala Dimbuene,

Thank you for submitting your manuscript to PLOS ONE. After careful consideration, we feel that it has merit but does not fully meet PLOS ONE’s publication criteria as it currently stands. Therefore, we invite you to submit a revised version of the manuscript that addresses the points raised during the review process.

We look forward to receiving your revised manuscript.

Kind regards,

Amos Buh, BSc., MPH, PhD

Academic Editor

PLOS ONE

Journal Requirements:

Reviewers' comments:

Reviewer's Responses to Questions

**Comments to the Author**

1. If the authors have adequately addressed your comments raised in a previous round of review and you feel that this manuscript is now acceptable for publication, you may indicate that here to bypass the “Comments to the Author” section, enter your conflict of interest statement in the “Confidential to Editor” section, and submit your "Accept" recommendation.

Reviewer #1: All comments have been addressed

Reviewer #2: All comments have been addressed

2. Is the manuscript technically sound, and do the data support the conclusions?

Reviewer #1: Yes

Reviewer #2: Yes

3. Has the statistical analysis been performed appropriately and rigorously? 

Reviewer #1: Yes

Reviewer #2: Yes

4. Have the authors made all data underlying the findings in their manuscript fully available?

Reviewer #1: Yes

Reviewer #2: Yes

5. Is the manuscript presented in an intelligible fashion and written in standard English?

Reviewer #1: Yes

Reviewer #2: Yes

6. Review Comments to the Author

Reviewer #1: As I indicated previously, the section “Analytical Strategy” from lines 261 through 269 is unnecessary and does not add useful depth or breadth to the manuscrpit. Moving it to an appendix would be sufficient if it must be included.

I thank the authors for adding the limitation section, but it remains rather short. Is the listed limitation the only one in this study?

This is an interesting area of research. Do the authors have suggestions for future research based on the results of this study?

Reviewer #2: All comments and modifications have been addressed

7. PLOS authors have the option to publish the peer review history of their article (what does this mean?). If published, this will include your full peer review and any attached files.

Reviewer #1: **Yes: **Dr. Ali M. AL-Asadi

Reviewer #2: No

---

## [Editor Report · Decision Letter 2]

5 Apr 2024

Men’s education and intimate partner violence—Beyond the victim-oriented perspective: Evidence from Demographic and Health Surveys in Central Africa

PONE-D-23-33735R2

Dear Dr. Tsala Dimbuene,

We’re pleased to inform you that your manuscript has been judged scientifically suitable for publication and will be formally accepted for publication once it meets all outstanding technical requirements.

Kind regards,

Amos Buh, BSc., MPH, PhD

Academic Editor

PLOS ONE